# A Novel and Adaptive Angle Diversity-Based Receiver for 6G Underground Mining VLC Systems

**DOI:** 10.3390/e24111507

**Published:** 2022-10-22

**Authors:** Pablo Palacios Játiva, Iván Sánchez, Ismael Soto, Cesar A. Azurdia-Meza, David Zabala-Blanco, Muhammad Ijaz, Ali Dehghan Firoozabadi, David Plets

**Affiliations:** 1Department of Electrical Engineering, Universidad de Chile, Santiago 8370451, Chile; 2Escuela de Informática y Telecomunicaciones, Universidad Diego Portales, Santiago 8370190, Chile; 3Department of Telecommunication Engineering, Universidad de Las Américas, Quito 170503, Ecuador; 4Department of Electrical Engineering, Universidad de Santiago de Chile, Santiago 9170124, Chile; 5Department of Computing and Industries, Universidad Católica del Maule, Talca 3466706, Chile; 6School of Engineering, Manchester Metropolitan University, Manchester M13 9PR, UK; 7Department of Electricity, Universidad Tecnológica Metropolitana, Av. Jose Pedro Alessandri 1242, Santiago 7800002, Chile; 8Department of Information Technology, Ghent University/IMEC, 9052 Ghent, Belgium

**Keywords:** Angle Diversity Receivers (ADRs), Underground Mining Visible Light Communication (UM-VLC), adaptive orientation receiver, Received Signal Strength Ratio (RSSR), VLC systems, 6G communication systems

## Abstract

Visible light communication (VLC) is considered an enabling technology for future 6G wireless systems. Among the many applications in which VLC systems are used, one of them is harsh environments such as Underground Mining (UM) tunnels. However, these environments are subject to degrading environmental and intrinsic challenges for optical links. Therefore, current research should focus on solutions to mitigate these problems and improve the performance of Underground Mining Visible Light Communication (UM-VLC) systems. In this context, this article presents a novel solution that involves an improvement to the Angle Diversity Receivers (ADRs) based on the adaptive orientation of the Photo-Diodes (PDs) in terms of the Received Signal Strength Ratio (RSSR) scheme. Specifically, this methodology is implemented in a hemidodecahedral ADR and evaluated in a simulated UM-VLC scenario. The performance of the proposed design is evaluated using metrics such as received power, user data rate, and bit error rate (BER). Furthermore, our approach is compared with state-of-the-art ADRs implemented with fixed PDs and with the Time of Arrival (ToA) reception method. An improvement of at least 60% in terms of the analyzed metrics compared to state-of-the-art solutions is obtained. Therefore, the numerical results demonstrate that the hemidodecahedral ADR, with adaptive orientation PDs, enhances the received optical signal. Furthermore, the proposed scheme improves the performance of the UM-VLC system due to its optimum adaptive angular positioning, which is completed according to the strongest optical received signal power. By improving the performance of the UM-VLC system, this novel method contributes to further consideration of VLC systems as potential and enabling technologies for future 6G deployments.

## 1. Introduction

Underground Mining (UM) environments poses a series of risks and safety issues that can affect the mine workers during their activities. These issues can be produced by the UM tunnels features, the potential toxic components, and high temperature, among other factors [1]. The factors that intervene in these scenarios require establishing reliable communication between the workers and the mining infrastructure in order to constantly verify their status and support critical applications for a correct mining operation. For example, localization systems in UM must be designed to work in real time, monitoring the well-being of each miner and allowing a fast response to any emergency [2]. Furthermore, the external factors of UM tunnels constitutes not just a rough environment for the workers but also an important difficulty to design communication systems. The electromagnetic interference, noise, and signal fading are factors that compromise the traditional communication systems. Consequently, a poor bit error rate (BER), delay spread, and limited rate of receiving data impact the operation of such systems [3]. A potential and feasible solution to mitigate these issues can be found in Visible Light Communication (VLC), which uses visible light to establish the communication link. In this context, and according to the progress and use of operating frequencies of 4G and 5G technologies, it is expected that 6G will operate with frequency bands above 50 GHz. Therefore, the use of VLC-based schemes could be a good option as part of the technologies to be deployed in 6G, since it operates in the range from 430 to 790 THz [4].

There are several advantages of VLC systems over Radio Frequency (RF) communication systems, which also justifies including VLC as part of the enabling technologies to be considered in the upcoming 6G standard. First, the spectrum is unlicensed and has a wide spectral range that would allow it to achieve high data transmission rates. Second, the light itself can be used as both illumination and communication, which achieves a reduction of costs while satisfying the underground mine norms. Third, the communication link is more secure because of the reduced coverage range. Despite these benefits, the environmental roughness of UM provokes several challenges from VLC systems. While the literature of VLC indoor systems is extensive [5], the UM-VLC systems do not consider assumptions that are normally used in VLC indoor systems, such as negligible scattering, perpendicular walls, orthogonal transmitters and receivers, or the lack of of shadowing due to mobile objects.

Among other problems identified in the literature in UM-VLC schemes are the low light coverage that exists in tunnels due to the low density of optical transmitters and the effect produced when the optical receivers are in the overlapping area of two adjacent optical cells [6]. These problems decrease the Signal-to-Interference-plus-Noise-Ratio (SINR) of the receivers considerably due to poor signal propagation and the existence of strong interference produced by the phenomenon of Inter-Cell Interference (ICI), which severely degrades the performance of the systems. Furthermore, specifically in UM environments, there are some SINR fluctuations, which are caused by Non-Line of Sight (non-LoS) components, which are due to reflections of the optical signal on irregular tunnel walls. This problem also reduces the system efficiency [7].

In the literature, there are several solutions to efficiently mitigate optical signal propagation issues. Recently, a particular technique, based on the Angle Diversity Receiver (ADR), has been introduced in indoor VLC environments [8]. An ADR consists of multiple narrow Field-of-View (FoV) Photo-Diodes (PDs) which, when combined, result in a large overall FoV and coverage as a single PD. Each of the narrow-FoV PDs can be selected or combined using signal combining schemes, thereby mitigating the effect of poor signal propagation and reducing the interference effect of ICI-producing signals. Another feature of these PD arrangements is to assume that the PDs orientation is fixed. However, limiting the PDs orientation also limits the flexibility of the applications and does not allow achieving better signal reception. Consequently, due to the hostility of the UM environments toward the VLC system, it is necessary to optimize the orientation and positioning of the PDs in the ADRs to improve the received signal and increase the system efficiency.

According to the orientation mechanisms reviewed in the literature, the orientation of PDs is commonly estimated by proximity, fingerprints [9], Time of Arrival (ToA) [10] and Received Signal Strength (RSS) mechanisms [11,12]. In this sense, algorithms based on RSS are the most used due to their high precision and low synchronization requirement. Specifically, one of these is the Received Signal Strength Ratio (RSSR) algorithm, which estimates the orientation of the target (the PD, in this case) depending on the received optical power. This parameter is affected by the relative distance and angles between the LED and the PD. However, the inherent challenge is to determine the influence of the relative angle between the LED and the PD in the RSSR method, since the PD orientation is commonly random and unknown in practice. Since the study of VLC systems applied to underground mines is considered a new area of application, these solutions have not been tested in these scenarios, either.

In the context of these challenges, in the next subsection, we will present and discuss the most relevant optical receiver orientation methods and algorithms in the state of the art and related works.

### Works Related to Receiver Orientation Algorithms and Methods

In general, receiving orientation algorithms are also applied for the location of receivers in a VLC scenario. For example, in [13], a hybrid RSS+Angle of Arrival (AoA) indoor positioning algorithm implemented in VLC is discussed. The interesting feature about this study is that ICI exists in the scenario, so the authors mitigate this problem using a unique frequency address. On the other hand, the authors in [14] present a mathematical model for a 3D positioning system in VLC. For this, they first derive the channel gain as a function of the source and receiver location in Cartesian coordinates. Then, they develop a cost function based on channel gain. In the experimental scenario, they consider receiver tilt, which is similar to UM-VLC environments. The results show that the proposed 3D method is more accurate compared to the 2D method. Along the same lines, the study carried out in [15] proposes a complete location and orientation solution for both the physical layer and the link layer of an indoor VLC system. The authors apply an RSS-based triangulation method for receptor location. In order to separate the optical signal in the receiver, the position of each LED is encoded with a unique location identification using Optical Orthogonal Codes (OOC). In this work, the impact of receiver orientation is also included. Although these works present certain slightly similar characteristics in the VLC scenario applied to underground mines, they do not consider all the variables of a mining tunnel. They also do not use RSS algorithms to orient receivers according to the best channel gain or power received but rather present the solutions to locate a receiver in the indoor scenario.

Under these precedents, few works have investigated the application of methods based on RSS or RSSR to efficiently target a PD or array of PDs and achieve better performance in the VLC system. In [16], the authors employ an array-based receiver of multidirectional PDs pointing in different directions and propose a 3D localization algorithm together with the array of PDs. This work uses the relative orientations of the PDs with respect to the receiver and the RSSR to mitigate the influence of the absolute orientation of the receiver. Although the work applies reception diversity to estimate PD positions, its focus is the location of the receiver, so it does not optimize its orientation. Furthermore, the applied scenario is an indoor VLC system. Other works only analyze the use of arrays of PDs called ADRs in UM environments in order to improve the reception of the optical signal in these scenarios, since it is highly affected by the hostility of the tunnel features and the physical phenomena that occur in the tunnel [6,17,18]. However, the ADRs keep the position and orientation of the PDs fixed. This solution alone, although it improves the quality of the reception, in certain locations of the UM tunnel requires mechanisms that automate the orientation of the PDs to obtain a better-received power. In [19], an in-depth analysis of VLC-based localization systems in which forward-facing algorithms are applied is presented. The authors present pioneering and more advanced articles specifically applied to VLC, where they are analyzed and classified according to localization techniques, types of transmitters and receivers, and multiplexing techniques. However, there is still the opportunity to apply these methods in UM-VLC systems. Finally, in [20], a high-coverage algorithm termed enhanced Camera-Assisted Received Signal Strength Ratio (eCA-RSSR) positioning algorithm is proposed for visible light positioning (VLP) systems. The basic idea of eCA-RSSR is to utilize visual information captured by the camera to estimate first the incidence angles of visible lights. Based on the incidence angles, eCA-RSSR utilizes the Received Signal Strength Ratio (RSSR) calculated by the photodiode (PD) to estimate the ratios of the distances between the LEDs and the receiver. The results of the work show that applications based on RSSR implemented in VLC systems have great acceptance and potential application for other environments.

After reviewing the state of the art and analyzing the opportunities for improving solutions in optical signal reception in UM environments, in this article, we propose an adaptive orientation receiver based on an ADR and RSSR orientation scheme adapted to an UM-VLC system with a view to establishing itself as a technology part of the 6G standard. As a base case, we use the hemidodecahedral ADR as a receiver along with the RSSR algorithm in the ADR to obtain the estimate of the optimal angle of incidence and orient the ADR in that direction. The proposed method consists of two phases: the reception of the optical signal by the PDs in the ADR to be compared between them and the estimation of the angle of incidence from the RSSR. The proposed solution allows one to improve the performance of the UM-VLC system without the limitation of the knowledge of the receiver orientation with emphasis on the best efficiency of the basic metrics for the evaluation of 6G technologies. Indeed, the proposed solution is evaluated in an UM scenario through computational simulations. Furthermore, it is compared with the pyramidal and hemidodecahedral ADRs of fixed PDs and with the ToA method through the received power, BER, and user data rate metrics. Therefore, the main contributions of this work are presented below:A theoretical analysis and design of an ADR structure with adaptive orientation is proposed in this work. Furthermore, the methodology to estimate the optimum angle of incidence on the receiver, along with its main mathematical expressions, is derived.A feasible and practical solution to improve the reception of the optical signal based on the adaptive orientation of the PDs in an ADR using an incidence angle estimation method through the RSSR tool.The evaluation and performance comparison of the presented solution with other typical receiver structures in an UM-VLC scenario in terms of important possible metrics for 6G technologies; such as received power, BER, and user data rate.

The remainder of this article is organized as follows. In Section 2, a brief contextualization of the UM-VLC channel model used is presented. In Section 3, an analysis of the adaptive orientation receiver, its scenario, and the incidence angle estimation method are presented. In Section 4, an evaluation of the solution and its comparison with other receivers in the UM-VLC scenario based on typical communication metrics are presented and discussed. Finally, relevant conclusions are presented in Section 5.

## 2. VLC System Model Applied to Underground Mining Environments

VLC systems applied to underground mines have marked differences compared to VLC systems applied to non-mining indoor environments. Among the features included in the modeling for UM-VLC systems are the angular positioning of LEDs and PDs, non-regular walls, and scattering and shadowing phenomena. Therefore, based on the work developed in the mining channel quote, in this section, we briefly describe the components of the VLC system applied to the underground mine, its components and characteristics.

### 2.1. Optical Transmitters and Receivers

As is widely known, in VLC systems, we assume that LEDs are a point source of light that follows a Lambertian pattern of radiation. In terms of position and orientation, LEDs in underground mines do not mount directly to the ceiling or point vertically downward. Therefore, in our work, we consider a set of *I* LEDs as optical transmitters. Furthermore, the physical position and the normal vector of an LED *i*, where i=1,2,…,I, are denoted by $\left(x_{i}^{T},y_{i}^{T},z_{i}^{T}\right)$ and $n_{i}^{tilt}$, respectively, as we can see in Figure 1.

On the other hand, we assume PDs as optical receptors, which will be located in an ADR that we will define in Section 3. The PDs are composed of a non-imaging concentrator (lens) and a physical active area $A_{p}$. In terms of position and orientation, when we install a PD in a mine worker’s helmet, it does not always point vertically upwards due to the very nature of the worker’s movement. Furthermore, the physical position and the normal vector of an PD *j*, where j=1,2,…,J, are denoted by $\left(x_{j}^{R},y_{j}^{R},z_{j}^{R}\right)$ and $n_{j}^{tilt}$, respectively, as we can see in Figure 1.

We assume that all LEDs have the same generalized Lambertian radiation pattern; therefore, the radiation intensity pattern $Ri(\phi_{ij}^{tilt})$ and the term $\cos(\phi_{ij}^{tilt})$ can be modeled as follows [21]:(1)$$Ri\left(\phi_{ij}^{tilt}\right)=\left\{\begin{array}{cc} \frac{m+1}{2\pi }\cos^{m}\left(\phi_{ij}^{tilt}\right) & \mathrm{if}\;-\pi /2\leq \phi_{ij}^{tilt}\leq \pi /2\\ 0 & \;\mathrm{otherwise} \end{array}\right.,$$
(2)$$\cos\left(\phi_{ij}^{tilt}\right)=\frac{V_{i-j}\cdot n_{i}^{tilt}}{∥V_{i-j}∥∥n_{i}^{tilt}∥},$$
where $\phi_{ij}^{tilt}$ is the radiance angle with respect to the normal vector to the LED *i*, the vector from LED *i* to PD *j* is denoted by $V_{i-j}$, the notation ∥·∥ denotes the two-norm, and · represents the product dot operation. In addition, for vector concepts, $∥V_{i-j}∥$ = $d_{ij}$, where $d_{ij}$ is the Euclidean distance between the LED *i* and PD *j*, $∥n_{i}^{tilt}∥$ = 1, and $n_{i}^{tilt}$ can be represented in terms of $\alpha_{i}$ and $\beta_{i}$ as $n_{i}^{tilt}=\left[\sin\left(\beta_{i}\right)\cos\left(\alpha_{i}\right),\sin\left(\beta_{i}\right)\sin\left(\alpha_{i}\right),-\cos\left(\beta_{i}\right)\right]$, where $\beta_{i}$ is the tilt angle with respect to the z-axis, which takes values in range of $\left[90^{\circ },180^{\circ }\right)$ and $\alpha_{i}$ is the rotation angle with respect to the x-axis, which is defined in the interval $\left[0^{\circ },360^{\circ }\right)$. Furthermore, $V_{i-j}=\left[x_{j}^{R}-x_{i}^{T},y_{j}^{R}-y_{i}^{T},-\Delta h_{ij}\right]$, assuming that $\Delta h_{ij}$ is the difference height between LED *i* and PD *j*, that is $z_{i}^{T}$ − $z_{j}^{R}=\Delta h_{ij}$.

In the receptor side, the effective collection area of the PD *j* and the term $\cos(\theta_{ij}^{tilt})$ acquire the form of [21]
(3)$$A_{eff}\left(\theta_{ij}\right)=\left\{\begin{array}{cc} \;A_{p}cos\left(\theta_{ij}^{tilt}\right) & \mathrm{if}\;-\Theta /2\leq \theta_{ij}^{tilt}\leq \Theta /2\\ 0 & \;\mathrm{otherwise} \end{array}\right.,$$
(4)$$\cos\left(\theta_{ij}^{tilt}\right)=\frac{V_{j-i}\cdot n_{j}^{tilt}}{∥V_{j-i}∥∥n_{j}^{tilt}∥},$$
where $\theta_{ij}^{tilt}$ is the incidence angle with respect to the normal vector to the PD *j*, $V_{j-i}$ is the vector from PD *j* to LED *i*, $∥V_{j-i}∥$ = $d_{ij}$, $∥n_{j}^{tilt}∥$ = 1, $n_{j}^{tilt}=\left[\sin\left(\beta_{j}\right)\cos\left(\alpha_{j}\right),\sin\left(\beta_{j}\right)\sin\left(\alpha_{j}\right),\cos\left(\beta_{j}\right)\right]$, where $\beta_{j}$ is the tilt angle with respect to the z-axis, which takes values in range of $\left[0^{\circ },90^{\circ }\right)$, and $\alpha_{j}$ is the rotation angle with respect to the x-axis that can be values in range of $\left[0^{\circ },180^{\circ }\right)$, $V_{j-i}=\left[x_{i}^{T}-x_{j}^{R},y_{i}^{T}-y_{j}^{R},\Delta h_{ij}\right]$, and Θ is the FoV of the PD. The gain of the optical concentrator can be written as $g\left(\theta_{ij}\right)=\eta^{2}/sin^{2}\left(\Theta \right)$, in which η is the internal refractive index of the concentrator.

### 2.2. Channel DC Gain

Based on the work completed in [21], the UM-VLC channel is modeled by integrating three optical components: the LoS component, the non-LoS component, and the scattering component. The LoS component is obtained directly from the LED lighting that falls on the PD. Thus, the LoS link depends on the LED and PD parameters as seen above. In addition, due to the UM infrastructure, in which large machinery and vehicles move through the tunnels, the effect of shadowing within the expression of the UM-VLC channel is considered. This UM scenario with its components is outlined in Figure 1. By integrating these factors, the Direct Current (DC) gain of the LoS optical wireless channel is formulated as follows:(5)$$\begin{array}{cc} H_{LoS,i,j} & =\frac{\left(m+1\right)A_{p}}{2\pi d_{ij}^{m+3}}{\left\{\left[x_{j}^{R}-x_{i}^{T},y_{j}^{R}-y_{i}^{T},-\Delta h_{ij}\right]\cdot \left[\sin\left(\beta_{i}\right)\cos\left(\alpha_{i}\right),\sin\left(\beta_{i}\right)\sin\left(\alpha_{i}\right),-\cos\left(\beta_{i}\right)\right]\right\}}^{m}\\ \; & \times \left\{\left[x_{i}^{T}-x_{j}^{R},y_{i}^{T}-y_{j}^{R},\Delta h_{ij}\right]\cdot \left[\sin\left(\beta_{j}\right)\cos\left(\alpha_{j}\right),\sin\left(\beta_{j}\right)\sin\left(\alpha_{j}\right),\cos\left(\beta_{j}\right)\right]\right\}G\left(\theta_{ij}^{tilt}\right)rect\left(\frac{\theta_{ij}^{tilt}}{\Theta }\right)P_{ij}. \end{array}$$
where $rect(\frac{\theta_{ij}^{tilt}}{\Theta })=1$ for $0\leq \theta_{ij}^{tilt}\leq \Theta$ and 0 otherwise, $G(\theta_{ij})$=$T_{s}\left(\theta_{ij}\right)$$g(\theta_{ij})$ represents the combined gain of the optical filter and optical concentrator, respectively, and $P_{ij}$ is a weighting function introduced to consider the random shadowing, which describes the probability that the LoS optical link is not blocked.

On the other hand, as a result of obstacles, and the non-flat and irregular walls of the tunnels, a diffuse component of the transmitted light is reflected by these elements. This effect generates the non-LoS component of the VLC channel, which is termed as $H_{NLoS}$. A common model for diffuse reflection is the Lambertian reflectance where light is reflected with equal radiance in all directions. However, this non-LoS component differs from the non-LoS component of non-mining indoor scenarios. The main differences are in the normal vectors of the non-planar walls, which are not orthogonal to the reflective area of the wall, which is called *w*. Therefore, these normal vectors ($n_{w}^{tilt}$) can be described in terms of the tilt angle with respect to the z-axis denoted as $\beta_{w}$, which belongs to the range of $\left[0^{\circ },180^{\circ }\right)$ and the rotation angle with respect to the x-axis denoted as $\alpha_{w}$ and takes values in range of $\left[0^{\circ },180^{\circ }\right)$.

The incidence angle with respect to the normal vector to the reflective element *w* and the radiance angle of the light component reaching the reflective element *w* are symbolized with $\theta_{iw}^{tilt}$ and $\phi_{wj}^{tilt}$, respectively. The angles of incidence and radiance are denoted by $\theta_{wj}^{tilt}$ and $\phi_{wj}$, respectively, which are measured with respect to the light component that is reflected in the reflective element *w* and reaches PD *j*. The effect of the non-flat walls is noticeable in terms of the following cosines: $\cos(\phi_{iw}^{tilt})$, $\cos(\theta_{iw}^{tilt})$, $\cos(\phi_{wj}^{tilt})$ and $\cos(\theta_{wj}^{tilt})$ in the following form:(6)$$\cos\left(\phi_{iw}^{tilt}\right)=\frac{V_{i-w}\cdot n_{i}^{tilt}}{∥V_{i-w}∥∥n_{i}^{tilt}∥},$$
here, $V_{i-w}$ is the vector from LED *i* to *w*, $∥V_{i-w}∥=d_{iw}$, where $d_{iw}$ is the Euclidean distances between LED *i* and the reflective element *w*, and $V_{i-w}=\left[x_{w}^{S}-x_{i}^{T},y_{w}^{S}-y_{i}^{T},-\Delta h_{iw}\right]$, assuming that the position of *w* is $\left(x_{w}^{S},y_{w}^{S},z_{w}^{S}\right)$, and $\Delta h_{iw}$ is the difference height between LED *i* and *w*.
(7)$$\cos\left(\theta_{iw}^{tilt}\right)=\frac{V_{w-i}\cdot n_{w}^{tilt}}{∥V_{w-i}∥∥n_{w}^{tilt}∥},$$
where $V_{w-i}$ is the vector from *w* to LED *i*, $∥V_{w-i}∥=d_{iw}$, $∥n_{w}^{tilt}∥$ = 1, $n_{w}^{tilt}$ can be represented in terms of $\alpha_{w}$ and $\beta_{w}$, that is $n_{w}^{tilt}=\left[\sin\left(\beta_{w}\right)\cos\left(\alpha_{w}\right),\sin\left(\beta_{w}\right)\sin\left(\alpha_{w}\right),\cos\left(\beta_{w}\right)\right]$ and $V_{w-i}=\left[x_{i}^{T}-x_{w}^{S},y_{i}^{T}-y_{w}^{S},\Delta h_{iw}\right]$.
(8)$$\cos\left(\phi_{wj}^{tilt}\right)=\frac{V_{w-j}\cdot n_{w}^{tilt}}{∥V_{w-j}∥∥n_{w}^{tilt}∥},$$
where $V_{w-j}$ is the vector from *w* to PD *j*, $∥V_{w-j}∥$ = $d_{wj}$, where $d_{wj}$ is the Euclidean distance between reflective element *w* and PD *j*, and $V_{w-j}=\left[x_{j}^{R}-x_{w}^{S},y_{j}^{R}-y_{w}^{S},-\Delta h_{wj}\right]$, assuming that $\Delta h_{wj}$ is the difference height between *w* and PD *j*.
(9)$$\cos\left(\theta_{wj}^{tilt}\right)=\frac{V_{j-w}\cdot n_{j}^{tilt}}{∥V_{j-w}∥∥n_{j}^{tilt}∥},$$
where $V_{j-w}$ is the vector from PD *j* to *w*, $∥V_{j-w}∥=d_{wj}$, and $V_{j-w}=\left[x_{w}^{R}-x_{j}^{R},y_{w}^{S}-y_{j}^{R},\Delta h_{wj}\right]$.

By including the effect of non-regular walls in the UM-VLC channel model and also consider the shadowing effect, the DC gain of the non-LoS optical wireless channel can be calculated by adding all the components arriving at the PD *j* after being reflected in a surface, namely
(10)$$\begin{array}{ll} H_{NLoS,i,j}^{\left(1\right)} & =\frac{\left(m+1\right)A_{p}}{2\pi }\sum_{w=1}^{W}\frac{\Delta A_{w}\rho_{w}}{d_{iw}^{m+3}d_{wj}^{4}}{\left\{\left[x_{w}^{S}-x_{i}^{T},y_{w}^{S}-y_{i}^{T},-\Delta h_{iw}\right]\cdot \left[\sin\left(\beta_{i}\right)\cos\left(\alpha_{i}\right),\sin\left(\beta_{i}\right)\sin\left(\alpha_{i}\right),-\cos\left(\beta_{i}\right)\right]\right\}}^{m}\\ & \times \left\{\left[x_{i}^{T}-x_{w}^{S},y_{i}^{T}-y_{w}^{S},\Delta h_{iw}\right]\cdot \left[\sin\left(\beta_{w}\right)\cos\left(\alpha_{w}\right),\sin\left(\beta_{w}\right)\sin\left(\alpha_{w}\right),\cos\left(\beta_{w}\right)\right]\right\}\\ & \times \left\{\left[x_{j}^{R}-x_{w}^{S},y_{j}^{R}-y_{w}^{S},-\Delta h_{wj}\right]\cdot \left[\sin\left(\beta_{w}\right)\cos\left(\alpha_{w}\right),\sin\left(\beta_{w}\right)\sin\left(\alpha_{w}\right),\cos\left(\beta_{w}\right)\right]\right\}\\ & \times \left\{\left[x_{j}^{R}-x_{w}^{S},y_{j}^{R}-y_{w}^{S},-\Delta h_{wj}\right]\cdot \left[\sin\left(\beta_{j}\right)\cos\left(\alpha_{j}\right),\sin\left(\beta_{j}\right)\sin\left(\alpha_{j}\right),\cos\left(\beta_{j}\right)\right]\right\}G\left(\theta_{wj}^{tilt}\right)rect\left(\frac{\theta_{wj}^{tilt}}{\Theta }\right)P_{iw}P_{wj}, \end{array}$$
where $\Delta A_{w}$ denotes the $w^{th}$ area of the considered reflective element *w*, whose reflection coefficient is represented by $\rho_{w}$, and *W* is the total number of reflective elements considered in the scenario. In addition, $P_{iw}$ and $P_{wj}$ are the weighted functions that consider the shadowing effect. These functions are obtained through the same statistical process to obtain $P_{ij}$, and they represent possible blockages in the optical link between LED *i* and the reflective element *w* and the optical link between *w* and PD *j*, respectively.

The last optical component that is included in the UM-VLC channel is the one produced by scattering. This phenomenon is caused by dust particles generated by the work of an underground mine. Therefore, its effect on the UM-VLC channel is direct in terms of magnitude and temporal dispersion. Based on the mathematical expression of the typical Lambertian channel model, the DC gain of the scattering optical wireless channel produced by scattering on the optical path LED *i*-$S_{n}$-PD *j*, which corresponds to the light beam that travels from the LED *i*, interacts with the local scatterrer $S_{n}$ and reaches the PD *j*, which can be written as [21]
(11)$$H_{sca,i,j}=\lim_{N\to \infty }\sum_{N=1}^{n}\frac{A_{p}\left(m+1\right)G_{n}\left(\mu \right)}{2\pi D_{i-n-j}^{2}}\cos^{m}\left(\phi_{i-S_{n}}\right)\cos\left(\theta_{S_{n}-j}\right)rect\left(\frac{\theta_{S_{n}-j}}{\Theta }\right),$$
where $G_{n}\left(\mu \right)$ is a coefficient introduced by each scatterer $S_{n}$, $\theta_{S_{n}-j}$ represents the angle between the vector from the $n^{th}$ scatterer of a set of *N* scatterers to PD *j* and $n_{j}^{tilt}$, $\phi_{i-S_{n}}$ is the radiance angle measured between $n_{i}^{tilt}$ and the vector from LED *i* to PD *j*, and the path length $D_{i-n-j}$ represents the total distance that light travels from the LED *i* via $S_{n}$ to the PD *j*. This path length can be expressed as $D_{i-n-j}=d_{i-S_{n}}+r_{n}$, where $d_{i-S_{n}}$ is the distance of the transmitted optical link between the LED *i* and $S_{n}$ and $r_{n}$ is the distance from the $n^{th}$ local scatterer to PD *j*.

Finally, the overall DC channel gain for UM-VLC systems is the sum of the LoS, non-LoS and scattering components, namely
(12)$$H_{miner,i,j}=H_{LoS,i,j}+H_{NLoS,i,j}^{\left(1\right)}+H_{sca,i,j}.$$

### 2.3. Receiver Optical Power

For the UM-VLC system, we denote as $P_{t}$ the optical power transmitted by a single LED. For simplicity, we assume that all LEDs in set *I* transmit the same $P_{t}$. Therefore, the optical power received at PD *j* from LED *i* is expressed as
(13)$$P_{r,i,j}=R_{PD}P_{t}H_{miner,i,j}+N_{j},$$
where $P_{r,i,j}$ is the power received of PD *j*, $R_{PD}$ is the PD responsivity, and $N_{j}$ is the additive noise in PD *j* that includes two types of noise that particularly affect UM environments, shot noise and thermal noise whose variances are denoted as $\sigma_{shot}^{2}$ and $\sigma_{thermal}^{2}$ respectively.

As mentioned in the contributions of the work, one of the objectives of this research is to study the impact of an adaptive orientation receiver applied to the presented UM-VLC system. As such, we will focus on a proposal which will allow us to obtain better system performance in terms of the power received at the receiver through the RSSR parameter. This proposal is based on an array of multidirectional PDs, which exploits the angular diversity of the array and the random orientation of the receiver located on the top of the mining workers’ helmets. The following section presents a rigorous mathematical analysis of the proposed receptor structure.

## 3. Adaptive Orientation Receiver

In an effort to improve the performance of VLC systems applied to underground mines and inspired by the characteristics of strong light directionality required in VLC, we propose a solution based on a adaptive orientation receiver.

### 3.1. Adaptive Orientation Receiver Structure

Basically, an adaptive orientation receptor consists of multiple PDs connected to a single signal processing chain, as shown in Figure 2. We can see that this receiver structure only requires a single Trans-Impedance Amplifier (TIA), which considerably reduces energy consumption in the optical signal reception and processing stage. In order to present a generalized model, we consider the set of PDs *J* defined in Section 2.1. For illustrative and demonstrative purposes, these PDs are installed in a hemidodecahedron geometric structure, which is described in detail in Section 4. However, our solution can be applied to any geometric structure (pyramidal, hemispherical, etc). Therefore, initially, the PDs point in different directions according to their position in the receiver of the hemidodecahedral structure that is installed in the helmet of the mining worker, as we can see in Figure 2.

For ease, we assume that all PDs in the receiver have the same optical characteristics, except for their position and pointing direction. In practice, even though the geometric structure is physically small, there are unavoidable differences between the positions of the PDs in the receiver. This causes variations between the distance from the LEDs to each PD. However, the distances among PDs are significantly small compared to the distance from the LEDs to the receiving geometric structure, so they are not considered. In addition, we consider that the orientation that the receiver takes will be based on the received power of the LoS component of the total UM DC channel gain, because it contributes with the largest magnitude optical power.

We denote the location of the hemidodecahedric structure as $\left(x^{H},y^{H},z^{H}\right)$, which according to the assumptions considered will be the common location for all PDs in the receiving structure. Consequently, we can affirm that $\left(x^{H},y^{H},z^{H}\right)$ = $\left(x_{j}^{R},y_{j}^{R},z_{j}^{R}\right)$∀j∈[1,J], $V_{i-j}$≈$V_{i}$ = $\left(x_{i}^{T},y_{i}^{T},z_{i}^{T}\right)$-$\left(x^{H},y^{H},z^{H}\right)$, $∥V_{i-j}∥$ = $d_{ij}$ ≈ $d_{i}$, where $d_{i}$ is the Euclidean distance from LED *i* to the receiving structure. Based on these mathematical considerations, we can approximately rewrite and expand expression (Equation 13) as
(14)$$\begin{array}{cc} P_{r,i,j} & \approx R_{PD}P_{t}H_{miner,i,j}+N_{j}\\ \; & \approx R_{PD}P_{t}\left[H_{LoS,i,j}+H_{NLoS,i,j}^{\left(1\right)}+H_{sca,i,j}\right]+N_{j}\\ \; & \approx R_{PD}P_{t}H_{LoS,i,j}+R_{PD}P_{t}H_{NLoS,i,j}^{\left(1\right)}+R_{PD}P_{t}H_{sca,i,j}+N_{j}\\ \; & \approx R_{PD}P_{t}\frac{\left(m+1\right)A_{p}}{2\pi d_{i}^{m+3}}{\left(V_{i}\cdot n_{i}^{tilt}\right)}^{m}\left(V_{i}\cdot n_{j}^{tilt}\right)G\left(\theta_{ij}^{tilt}\right)P_{i}+R_{PD}P_{t}H_{NLoS,i}^{\left(1\right)}+R_{PD}P_{t}H_{sca,i}+N_{H}\\ \; & \approx \frac{C}{d_{i}^{m+3}}{\left(V_{i}\cdot n_{i}^{tilt}\right)}^{m}\left(V_{i}\cdot n_{j}^{tilt}\right)P_{i}+R_{PD}P_{t}H_{NLoS,i}^{\left(1\right)}+R_{PD}P_{t}H_{sca,i}+N_{H}, \end{array}$$
where *C*=$\frac{\left(m+1\right)A_{p}}{2\pi }R_{PD}P_{t}G\left(\theta_{ij}^{tilt}\right)$ is a constant, $P_{i}$ is the weighing function that describes the probability that the LoS link between the LED *i* and the entire receiving structure is blocked, $H_{NLoS,i}^{\left(1\right)}$ and $H_{sca,i}+N_{H}$ are the non-LoS channel and scattering components between the LED *i* and the receiver structure, respectively, and $N_{H}$ is the additive noise in the receiver structure.

As mentioned in this section, we will focus mainly on the UM LoS channel component, i.e., the former component of $P_{r,i,j}$. Therefore, our analysis and algorithm that we propose below is based on that assumption.

### 3.2. Adaptive Receptor Orientation Scenario

In the proposed algorithm, we assume that there are at least two LEDs within the hemidodecahedral geometric arrangement FoV, which consists of six PDs, five on its lateral faces and one on its upper face, as we can see in Figure 2. However, to generalize with any geometric arrangement, we need at least three PDs, that is, *I*≥ 2 and *J*≥ 3. We also consider that the FOV of the receiving structure is the intersection of the FoV of all the PDs that compose it. We also assume that each LED takes turns transmitting its $\left(x_{i}^{T},y_{i}^{T},z_{i}^{T}\right)$ and $n_{i}^{tilt}$ in its assigned time slot through visible light; when the optical signal emitted by the LED is within the FoV of the PD, the PD receives the signal and also measures the corresponding received signal power.

As we noted in expression (Equation 14), the position, rotation and tilt of the LEDs and PDs directly affect the UM-VLC channel and therefore the received power. However, since the LEDs are fixedly installed, we will focus on the positioning of the PDs in the receiver structure. This positioning affects the incidence angle, as we observe in expression (Equation 4), with the $\alpha_{j}$ and $\beta_{j}$ angles. Therefore, in the algorithm that we propose, we will first estimate the $\theta_{ij}^{tilt}$, to then obtain the $\alpha_{j}$ and $\beta_{j}$ values, and finally estimate the orientation that the receiving structure must follow.

As illustrated in Figure 1, we employ a Cartesian coordinate system with respect to the tunnel to specify positions of the elements of the system. On the other hand, to estimate $\theta_{ij}^{tilt}$, we need to know the relationships among the different orientations that PDs can take in the receiving structure. However, these orientations are random and unknown if based on the coordinate system with respect to the tunnel. This problem occurs because the receiving structure can rotate and tilt randomly by the movement of the mining worker’s head. In contrast, PDs are initially fixed with respect to the hemidodecahedral receptor structure. Thus, it is necessary to introduce a secondary Cartesian coordinate system with respect to the receiving structure to specify the PDs orientations.

Based on the system distribution in Figure 1 and Figure 2, the origin point of the secondary coordinate system, $O^{H}$, is the center of the hemidodecahedron. The $x^{H}$−$y^{H}$ plane coincides with the bottom plane of the hemidodecahedron and the positive axis $x^{H}$ coincides with the normal direction of the hemidodecahedron. By convention, the inclination angles of the PDs in the geometric structure start from zero. To distinguish and differentiate the vector and position notations with respect to the main coordinate system, in the secondary coordinate system, we use the superscript ${"}^{H}"$ for the respective representation. Therefore, the positions $\left(x_{i}^{T},y_{i}^{T},z_{i}^{T}\right)$, $\left(x_{j}^{R},y_{j}^{R},z_{j}^{R}\right)$, and $\left(x^{H},y^{H},z^{H}\right)$, and the vectors $n_{i}^{tilt}$, $n_{j}^{tilt}$, and $d_{ij}$ in the main coordinate system are represents by the positions $\left(x_{i}^{H},y_{i}^{H},z_{i}^{H}\right)$, $\left(x_{j}^{H},y_{j}^{H},z_{j}^{H}\right)$, and $\left(x^{H},y^{H},z^{H}\right)$, and the vectors $n_{i}^{tilt(H)}$, $n_{j}^{tilt(H)}$, and $d_{ij}^{H}$ in the secondary coordinate system.

Considering these details and based on the receptor hemidodecahedron geometry, $\left(x^{H},y^{H},z^{H}\right)$=(0,0,0), $\left(x_{j}^{H},y_{j}^{H},z_{j}^{H}\right)$ and $n_{j}^{tilt(H)}$ are represented as
(15)$$\begin{array}{c} \left(x_{j}^{H},y_{j}^{H},z_{j}^{H}\right)=\left(ap\left[1+cos(\alpha_{j}^{H})\right]cos\left(\beta_{j}^{H}\right),ap\left[1+cos(\alpha_{j}^{H})\right]\sin\left(\beta_{j}^{H}\right),\frac{ap\left[1+cos(\alpha_{j}^{H})\right]}{tan(\alpha_{j}^{H})}\right), \end{array}$$
(16)$$n_{j}^{tilt(H)}=\left[\sin\left(\beta_{j}^{H}\right)\cos\left(\alpha_{j}^{H}\right),\sin\left(\beta_{j}^{H}\right)\sin\left(\alpha_{j}^{H}\right),\cos\left(\beta_{j}^{H}\right)\right],$$
(17)$$\alpha_{j}^{H}=\left\{\begin{array}{cc} 0^{\circ }, & \mathrm{if}\;j=1\\ \alpha_{j}, & \mathrm{if}\;j=2,\ldots ,6, \end{array}\right.$$
(18)$$\beta_{j}^{H}=\frac{2\pi (j-1)}{5}\;\;\;\;\;\;\;\;for\;\;2\leq j\leq 6,$$
where ap is the apothem of the hemidodecahedron and is represented by $ap=\frac{L}{2tan(\omega /2)}$, with *L* as the length of the sides of the hemidodecahedron lateral surfaces and *w* as its central angle. The angles $\alpha_{j}^{H}$ and $\beta_{j}^{H}$ are are the angles of rotation and tilt of each PD *j* with respect to the secondary coordinate system, respectively.

### 3.3. Methodology Used to Estimate the Angle of Incidence


To obtain the proper orientation that the receiving geometric structure will be given, we must have a fairly accurate estimate of the direction of the LoS optical link. Therefore, the suitable variable to estimate is the angle of incidence. In this research, we propose the estimation of $\theta_{ij}^{tilt}$ based on the RSSR metric implemented and analyzed from the reference of the secondary coordinate system.

According to the mathematical expression of received power derived in (Equation 14), we define in a general way that the approximate RSSR between PD *j* and any PD *q*, where *j* and *q* belong to the set of PDs *J* corresponding to the receiving geometric structure, is expressed as
(19)$$\begin{array}{cc} \frac{P_{r,i,j}}{P_{r,i,q}} & \approx \frac{\frac{C}{d_{i}^{m+3}}{\left(V_{i}\cdot n_{i}^{tilt}\right)}^{m}\left(V_{i}\cdot n_{j}^{tilt}\right)P_{i}+R_{PD}P_{t}H_{NLoS,i}^{\left(1\right)}+R_{PD}P_{t}H_{sca,i}+N_{H}}{\frac{C}{d_{i}^{m+3}}{\left(V_{i}\cdot n_{i}^{tilt}\right)}^{m}\left(V_{i}\cdot n_{q}^{tilt}\right)P_{i}+R_{PD}P_{t}H_{NLoS,i}^{\left(1\right)}+R_{PD}P_{t}H_{sca,i}+N_{H}}, \end{array}$$

In order to obtain simplified analytic expressions based on the direction cosines of the incidence angles between LED *i* and PDs *j* and *q* together with their normal vectors, and expressing them as a function of the Cartesian secondary coordinate system with respect to the receiver, we can rewrite Equation (Equation 19) as
(20)$$\begin{array}{cc} \frac{P_{r,i,j}}{P_{r,i,q}} & \approx \frac{\cos(\theta_{ij}^{tilt})}{\cos(\theta_{iq}^{tilt})}=\frac{n_{j}^{tilt(H)}\cdot d_{ij}^{H}}{n_{q}^{tilt(H)}\cdot d_{iq}^{H}}, \end{array}$$
where $d_{ij}^{H}$ and $d_{iq}^{H}$ can be considered approximately equal, since they would be almost the same distance from LED *i*. Therefore, we will define this general distance as $d_{i}^{H}$. As mentioned before, to generalize, we define *J* ≥ 3 PDs. Therefore, we can obtain for each LED in the system (*L* − 1) independent RSSR equations similar to expression (Equation 20). For simplicity in the calculations, we consider the RSSR between the first PD and the other PDs in the array to obtain the following expression:(21)$$\begin{array}{cc} \frac{P_{r,i,j}}{P_{r,i,1}} & \approx \frac{n_{j}^{tilt(H)}\cdot d_{i}^{H}}{n_{1}^{tilt(H)}\cdot d_{i}^{H}}. \end{array}$$

This expression can be restructured as
(22)$${\left(n_{j}^{tilt(H)}-\frac{P_{r,i,j}}{P_{r,i,1}}n_{1}^{tilt(H)}\right)}^{T}d_{i}^{H}=0.$$

Since the module of $d_{i}^{H}$ does not directly influence the equation, only the direction of of $d_{i}^{H}$ is needed. Therefore, we can define the independent RSSR set of equations in matrix form as
(23)$$A_{i}≜{\left[\begin{array}{c} {\left(n_{2}^{tilt(H)}-\frac{P_{r,i,2}}{P_{r,i,1}}n_{1}^{tilt(H)}\right)}^{T}\\ \vdots \\ {\left(n_{L}^{tilt(H)}-\frac{P_{r,i,L}}{P_{r,i,1}}n_{1}^{tilt(H)}\right)}^{T} \end{array}\right]}_{(L-1)\times 3},$$
(24)$$x_{i}≜d_{i}^{H}.$$

We can express these matrices as a group of homogeneous linear equations defined as
(25)$$A_{i}x_{i}=0;\;\;\;\;subject\;to\;||x_{i}\left|\right|=1.$$

Finally, we obtain the linear solution of (Equation 25) by applying the method of least squares $\left({\hat{x}}_{i}\right)$. This expression is given by the eigenvector corresponding to the smallest eigenvalue of $A_{i}^{T}A_{i}$. Furthermore, since $0\leq \theta_{ij}^{tilt}\leq \Theta$, we can infer that $\cos(\theta_{ij}^{tilt})\geq 0$. Therefore, the estimate incidence angle can be calculated as
(26)$$\hat{\theta_{ij}^{tilt}}=\arccos\left(|n_{j}^{tilt(H)}\hat{x_{i}}|\right).$$

With the estimate $\hat{\theta_{ij}^{tilt}}$ value and through mechanical rotation mechanisms, we can optimally orient the PD structure so that it can maximize the power received by the LED.

## 4. Results and Discussion

In this section, we simulate and present graphical results based on numerical simulations using the Monte Carlo method in Matlab. The findings obtained are based on evaluation metrics of traditional wireless communication systems. These parameters together with the application of the UM-VLC system allow us to examine the capacity, behavior, and performance of the adaptive orientation receiver solution presented. Furthermore, to validate and compare our proposal with typical state-of-the-art solutions, we make a fair comparison with pyramidal and hemidodecahedral ADRs with fixed PDs and with the ToA method. Among the metrics used to evaluate the presented solution are the power received and the user data rate distributed in the scenario together with the BER. Both the parameters of the simulation model that were used in this manuscript as well as the description of the UM-VLC scenario are presented in Table 1.

### 4.1. Distribution of the Received Power

Figure 3 shows the distribution of the power received by the hemidodecahedral ADR with PDs with adaptive orientation, with a fixed orientation, and with the ToA method as well as the distributions for pyramidal ADRs with fixed PDs and with the ToA method. These distributions are calculated based on Equations (Figure 2) and (Equation 21). We evaluate the received power by keeping the coordinate *z* of the ADRs with a value of 1.8 m because it is a typical position value of the communication devices in these scenarios. In addition, $P_{t}$ is set to 5 W to provide constant lighting in the UM tunnel.

In the first instance, we can see that by implementing the ToA method in the hemidodecahedral ADR, we improve the power received compared to the same ADR with fixed PDs by 20%. In addition, by implementing this method in the pyramidal ADR and comparing it with its pair with fixed PDs, we also improve the power received by 10%. However, if we compare them with our proposal and take as a precedent that the hemidodecahedral ADR by itself improves the efficiency of the received signal with respect to the pyramidal ADR, we can observe that the presented proposal (Figure 3a) presents the best performance in terms of the evaluated metric compared to its evaluative ADR peers. Indeed, we have power values between −45.3 and −81.2 dBm for the hemidecahedral ADR with PDs with adaptive orientation. This effect of improvement of the received power is due to the automatic orientation that the PDs have in the array. Therefore, wherever the ADR is in the UM-VLC scenario, it will always point its PDs in the direction that maximizes the power gain.

### 4.2. Distribution of the User Data Rate

The user data rate is a metric that allows us to validate the maximum and minimum data transfer values that are received by the ADRs. Figure 4 shows the UDR distribution for the proposed solution based on the adaptive orientation of PDs in the hemidodecahedral ADR comparing it with the ADRs with the ToA method and fixed PDs. We can clearly see that, as with the received power, the best system performance is achieved with the proposal presented in this article in terms of the UDR. As we can see in the Figure 4a, UDR values between 19.2 and 185.4 Mbps are obtained, having full coverage in the UM-VLC scenario. In Table 2, we can also verify the improvement percentage of our solution compared to state-of-the-art solutions. These results allow us to deduce and re-validate that the improvement in the system performance is due to the fact that, in addition to the quantity and distribution of PDs in the hemidodecahedral geometry, this ADR architecture is better used and optimized in conjunction with the adaptive orientation PDs methodology according to the better reception of the optical signal.

### 4.3. Bit Error Rate

An important metric that evaluates the complete UM-VLC system performance including the proposed solution is the BER. By using this metric, we can find the number of bits that have been altered either by the effects of the UM environment or by inefficiency in receiving solutions. This evaluation is carried out using Monte Carlo simulations and by considering the LED transmission based on the PHY-I mode with the On–Off Keying (OOK) modulation [7,28].

Figure 5 shows the BER curves obtained for the hemidodecahedral solution with adaptive orientation PDs and the ADRs of the state of the art in the proposed UM-VLC scenario. It is important to note that, like the other metrics previously analyzed, the curve that describes the BER of the solution presented in this article is the one that presents the best performance. Indeed, we can verify that when compared to the pyramidal ADR, its performance based on BER improves notably. This gain is due to the orientation and positioning efficiency of the PDs in the hemidodecahedral ADR due to the implemented methodology. This result would also allow us to deduce that the proposed solution mitigates to a certain extent the harmful effects of the UM-VLC environment, which drastically degraded the BER. Furthermore, given the automatic oriented granularity of the PDs in the ADR, the ICI produced is mitigated, since the interference caused by LEDs whose received power is not optimal is minimized.

## 5. Conclusions

This article presents a novel and adaptive angle diversity-based receiver for 6G UM-VLC systems. The performance of the adaptive orientation solution based on the RSSR scheme, which is implemented in a hemidodecahedral ADR, is verified. The architecture and methodology of the proposed solution were analyzed mathematically and geometrically in the reception stage of an UM-VLC scheme through computational simulations. The evaluation was performed in terms of received power, user data rate, and BER, and its numerical results were compared with the pyramidal and hemidecahedral ADRs solutions with fixed PDs and considering the ToA orientation method. The findings derived from the described metrics demonstrate that the solution proposed in this manuscript perform better than typical ADRs in UM-VLC schemes; consequently, it improves the fulfillment of the communication goals of the UM-VLC system. Specifically, for the user data rate distribution in the UM-VLC scenario, the solution proposed in this article presents an improvement of over 50% with respect to the state-of-the-art solutions used for comparison in this work. On the other hand, when we analyze the BER curves, the proposed solution enhances the BER of the system compared to typical ADR solutions. Therefore, the enhancement of the system is due to the mitigation of ICI, and the external factors that affect the UM-VLC environment due to the efficient, as well as adaptive and automatic orientation that the PDs have in the hemidodecahedral ADR. The improvement in signal reception performance implies an improvement in the operation of the UM-VLC system, which increases the chances of being considered as part of the enabling technologies for 6G wireless communications systems.

As future work, a physical and experimental implementation and validation of the solution presented in this manuscript will be considered along with electromechanical mechanisms that generate the automatic adaptive orientation of the PDs. Furthermore, the behavior and performance of the created solution will be verified in other VLC environments based on specific channel models for these environments.

## Figures and Tables

**Figure 1 entropy-24-01507-f001:**
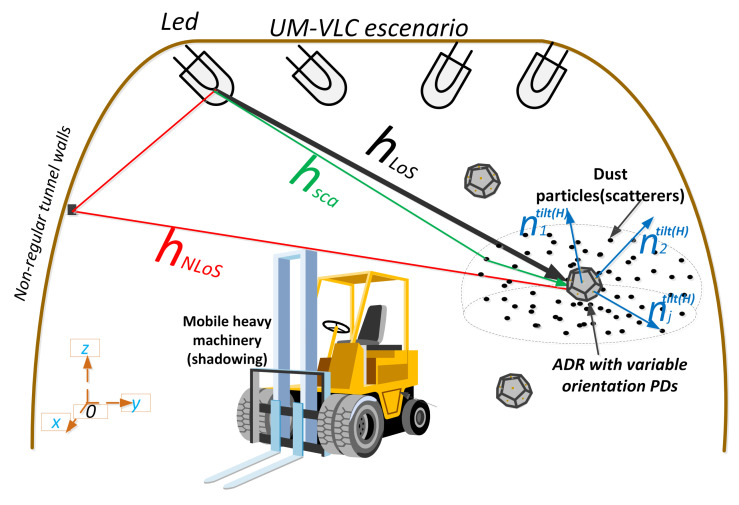
UM-VLC propagation scenario with the geometry of optical channel components.

**Figure 2 entropy-24-01507-f002:**
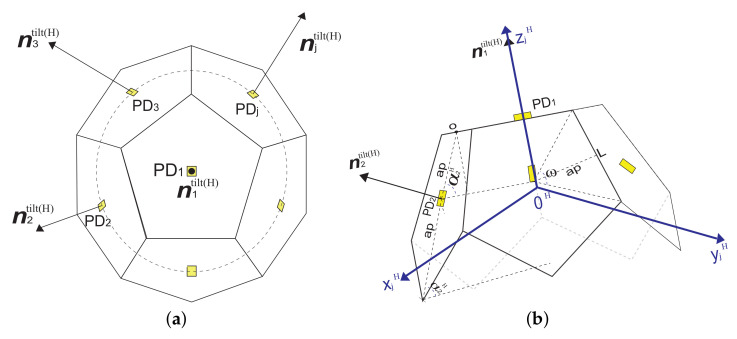
General structure of the hemi-dodecahedron ADR with adaptive orientation PDs. (**a**) Top view and (**b**) side view.

**Figure 3 entropy-24-01507-f003:**
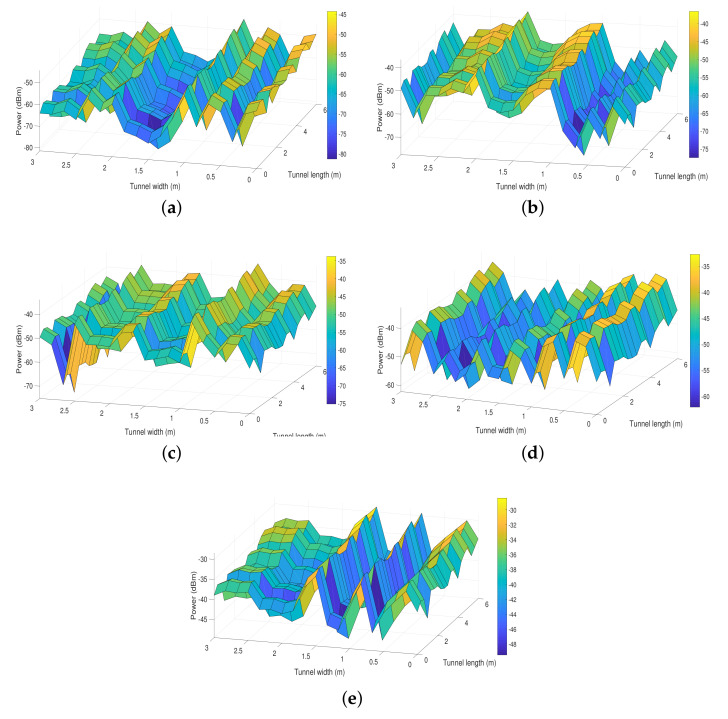
Distribution of the received power in the UM-VLC scenario with (**a**) hemidodecahedral ADR with PDs with adaptive orientation, (**b**) hemidodecahedral ADR with PDs fixed, (**c**) hemidodecahedral ADR with the ToA method, (**d**) pyramid ADR with the ToA method, and (**e**) pyramid ADR with the PDs fixed.

**Figure 4 entropy-24-01507-f004:**
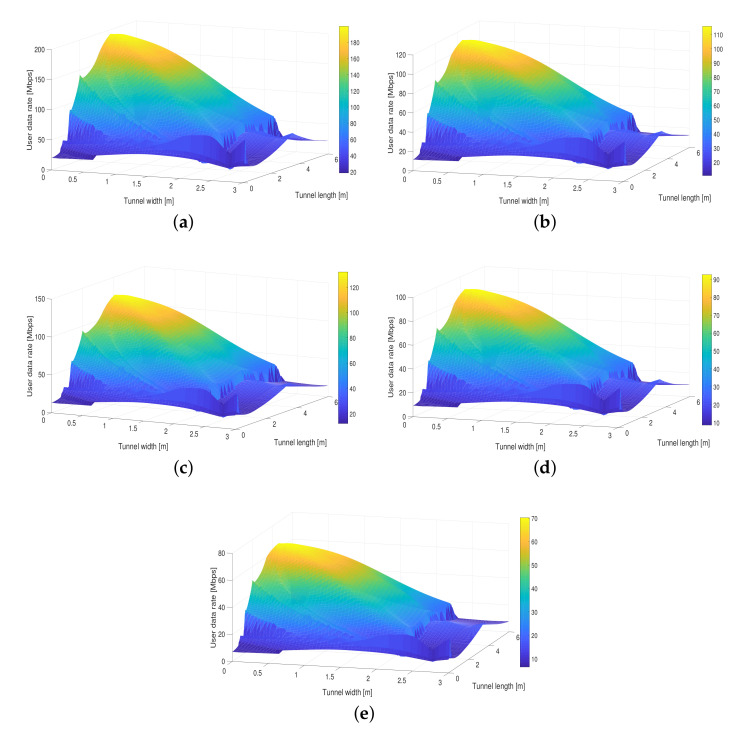
Distribution of the user data rate in the UM-VLC scenario with (**a**) hemidodecahedral ADR with adaptive orientation PDs, (**b**) hemidodecahedral ADR with PDs fixed, (**c**) hemidodecahedral ADR with the ToA method, (**d**) pyramid ADR with the ToA method, and (**e**) pyramid ADR with fixed PDs.

**Figure 5 entropy-24-01507-f005:**
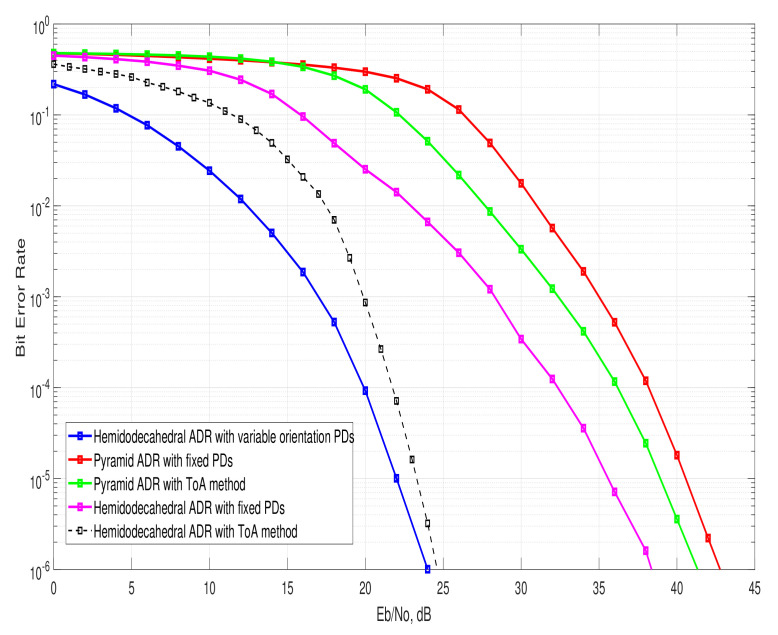
BER curves for the hemidodecahedral ADR with adaptive orientation PDs and the state-of-the-art ADRs architectures.

**Table 1 entropy-24-01507-t001:** UM-VLC system simulation parameters.

UM Simulation Scenario	Values	Ref.
**Tunnel Parameters**		
Dimensions (w×l×h), m	(3×6×5)	
Coordinates of the LED(x,y,z), m	$LED_{1}=\left(3,1.5,4.5\right)$, $LED_{2}=\left(3,3,4.5\right),$	
	$LED_{3}=\left(3,4.5,4.5\right),$ $LED_{4}=\left(-3,6,4.5\right)$	
**Channel and element parameters**		
AWGN power spectral density (A/Hz)	$2.5\times 10^{-23}$	[22]
LED rotation angle, ${(}^{\circ })$	45	[7]
LED tilt angle, ${(}^{\circ })$	45	[7]
Noise bandwidth (MHz)	100	[23]
Wall reflection coefficient, $\rho_{w}$	0.6	[7]
Wall rotation angle, ${(}^{\circ })$	*U*[0,180]	[7]
Wall tilt angle, ${(}^{\circ })$	*U*[0,180]	[7]
**VLC transceiver parameters**		
Average transmitted power, $P_{t}$ (W)	5	[24]
Band-pass filter of transmission	1	[6]
FoV, $\Theta \;{(}^{\circ })$	70	[24]
Gain of the optical filter	1	[24]
Lambertian mode number, *m*	1	[25,26]
LED wavelength, λ (nm)	580	[24]
Modulation type	OOK	[27]
Modulation bandwidth (MHz)	50	[27]
Modulation index	0.3	[27]
Optical filter bandwidth (nm)	340 to 694.3	[24]
Optical filter center wavelength (nm)	580 ± 2	[24]
Optical filter full width half max (nm)	10 ± 2	[24]
Physical active area, $A_{p}$ (cm^2^)	1	[6]
Reflective element area, $\Delta A_{w}\;\left({\mathrm{cm}}^{2}\right)$	1	[21]
Refractive index, $m_{s}$	1.5	[6]
Responsivity, $R_{PD}$ (A/W)	0.53	[6]
Semi-angle at half power, ${(}^{\circ })$	60	[24]

**Table 2 entropy-24-01507-t002:** Percentage improvement in the user data rate of the proposed solution compared to state-of-the-art solutions.

State-of-the-Art Solutions	Hemidodecahedral ADR with PDs with Adaptive Orientation
Hemidodecahedral ADR with PDs fixed	61.1%
Hemidodecahedral ADR with the ToA method	50.4%
Pyramid ADR with the ToA method	101.3%
Pyramid ADR with the PDs fixed	160.2%

## Data Availability

The data presented in this study are available on request from the corresponding author.

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
