# Peer review of "A Novel and Adaptive Angle Diversity-Based Receiver for 6G Underground Mining VLC Systems"

_entropy, 2022, doi:10.3390/e24111507_

Round 1

Reviewer 1 Report

The authors present a solution that improves the underground mining VLC (UM-VLC) systems by using an adaptive orientation receiver based on an angle diversity receiver (ADR) and the received signal strength ratio (RSSR) orientation scheme. The architecture and methodology of the solution were mathematically and geometrically analyzed through computational simulations. The performance of the system was evaluated by using the receiver power, user data rate, and bit error rate metrics, and the numerical results were compared with the state-of-the-art solutions. The comparison demonstrated that the proposed solution performed better than the typical ADR schemes, therefore improving the UM-VLC system. In terms of metrics, the user data rate distribution presented an improvement of 50% with respect to the state-of-the-art solutions, and the BER curves were also enhanced.

Minors: The treatment of the subject is insufficient. It is difficult for the reader to place the contribution of the work in the literature. This is also reflected by the small bibliography.

Consider redrawing the figures with bigger fonts and also making sure you don't break the aspect ratio, for a better presentation.

Overall, this is a good contribution in the field of underground mining.

Author Response

Thank you for your valuable comments. In the attached document you will find our response to your comments and suggestions.

Reviewer 2 Report

The article Novel and Adaptive Angle Diversity-based Receiver for 6G Underground Mining VLC Systems is focused on 6G VLC systems and their application in underground mining environment. The article presents a novel solution that involves an improvement to the Angle Diversity Receivers (ADRs) based on the adaptive orientation of the Photo-Diodes (PDs) in terms of the Received Signal Strength Ratio (RSSR) scheme based on hemidodecahedral ADR. The proposed scheme improves the performance of the UM-VLC system due to its optimum adaptive angular positioning, which is done according to the strongest optical received signal power.

First, the organization of the paper is acceptable, the paper contains both introduction as well as theoretical and simulation sections supported with the results and conclusions. The presentation of the results could be perhaps improved and more attention should be paid to the graphical presentation of the results.

The language throughout the entire paper is acceptable, however, I was able to notice several minor mistakes and errors, therefore I recommend to perform some proofreading and corrections. Generally, the language is technically sound.

I recommend to perform more extensive simulations and to include simulations for various different channel and environment scenarios in order to demonstrate the performance of the proposed solution for various different environments.

Author Response

(The authors gave the same response as above.)
